# Repetitively Administered Low-Dose Donor Lymphocyte Infusion for Prevention of Relapse after Allogeneic Stem Cell Transplantation in Patients with High-Risk Acute Leukemia

**DOI:** 10.3390/cancers13112699

**Published:** 2021-05-30

**Authors:** Panagiotis Tsirigotis, Konstantinos Gkirkas, Vassiliki Kitsiou, Spiros Chondropoulos, Theofilos Athanassiades, Thomas Thomopoulos, Alexandra Tsirogianni, Maria Stamouli, Aggeliki Karagiannidi, Nikolaos Siafakas, Vassiliki Pappa, Arnon Nagler

**Affiliations:** 1Second Department of Internal Medicine, ATTIKO General University Hospital, National and Kapodistrian University of Athens, 12462 Athens, Greece; kostasgirkas@gmail.com (K.G.); spiroshont@gmail.com (S.C.); ththomop@med.uoa.gr (T.T.); maristabmt@gmail.com (M.S.); angie.krg@gmail.com (A.K.); vaspappa@med.uoa.gr (V.P.); 2Department of Immunology and Histocompatibility, Evangelismos General Hospital, 10676 Athens, Greece; kiki.kitsiou@gmail.com (V.K.); theofilosathanassiades@gmail.com (T.A.); alextsir@gmail.com (A.T.); 3Clinical Microbiology Laboratory, ATTIKO General University Hospital, National and Kapodistrian University of Athens, 12462 Athens, Greece; nsiaf@med.uoa.gr; 4Chaim Cheba Medical Center, Hematology and Bone Marrow Transplantation Department, Tel-Hashomer 52621, Israel; Arnon.Nagler@sheba.health.gov.il

**Keywords:** prophylactic donor lymphocyte infusion, allogeneic stem cell transplantation, high-risk acute leukemia, prevention of relapse, graft versus host disease

## Abstract

**Simple Summary:**

Patients with high-risk acute leukemia (AL) have a high risk of relapse after allogeneic stem cell transplantation (allo-SCT). Prophylactic donor lymphocyte infusion (pro-DLI) has shown significant efficacy in reducing relapse rate but its widespread application has been limited due to concerns of graft versus host disease (GVHD) development. In the present study, we tested the safety and efficacy of a novel method of pro-DLI based on repetitive administration of low lymphocyte doses. Low-dose pro-DLI was administered every two months for at least 3-years. Forty-four high-risk AL patients with a median age of 53-years who underwent allo-SCT from a matched-sibling (38pts) or a matched-unrelated donor (6pts) received pro-DLI. Acute and chronic-GVHD developed in 14% and 12% of patients. With a median follow-up of 44-months, the cumulative incidence of relapse and non-relapse mortality was 12% and 13%, while the overall survival was 78%. Prolonged up to 3-years low-dose pro-DLI administered every two months is safe and effective in reducing relapse rate in patients with high-risk AL. The low-dose repetitive administration DLI strategy reduced the risk of DLI-mediated GVHD, while the prolonged repeated administration helped in preventing relapse, possibly by inducing a sustained and prolonged immunological pressure on residual leukemic cells.

**Abstract:**

Background: Patients with high-risk acute leukemia have a high risk of relapse after allogeneic stem cell transplantation (allo-SCT). In an effort to reduce the relapse rate, various therapeutic methods have been implemented into clinical practice. Among them, prophylactic donor lymphocyte infusion (pro-DLI) has shown significant efficacy. However, the widespread application of pro-DLI has been restricted mostly due to concerns regarding the development of graft versus host disease (GVHD). In the present study, we tested the safety and efficacy of a novel method of prophylactic-DLI based by repetitive administration of low lymphocyte doses. Methods: DLI was administered to patients with high-risk acute leukemia at a dose of 2 × 10^6^/kg CD3-positive cells. DLI at the same dose was repeated every two months for at least 36 months post-allo-SCT, or until relapse or any clinical or laboratory feature suggested GVHD, whichever occurred first. Forty-four patients with a median age of 53 years (range 20–67) who underwent allo-SCT between 2011 and 2020 were included in our study. Thirty-three patients with high-risk acute myeloid leukemia (AML) and 11 with high-risk acute lymphoblastic leukemia (ALL) after allo-SCT from a matched sibling (MSD, no = 38 pts) or a matched-unrelated donor (MUD, no = 6 pts) received pro-DLI. Twenty-three patients were in CR1, all with unfavorable genetic features; 12 patients were in CR2 or beyond; and 9 patients had refractory disease at the time of transplant. Ten out of 23 patients in CR1 had detectable minimal residual disease (MRD) at the time of allo-SCT. Disease risk index (DRI) was high and intermediate in 21 and 23 patients, respectively. Conditioning was myeloablative (MAC) in 36 and reduced intensity (RIC) in 8 patients, while GVHD prophylaxis consisted of cyclosporine-A in combination with low-dose alemtuzumab in 39 patients or with low-dose MTX in 5 patients, respectively. Results: Thirty-five patients completed the scheduled treatment and received a median of 8 DLI doses (range 1–35). Fifteen out of 35 patients received all planned doses, while DLI was discontinued in 20 patients. Reasons for discontinuation included GVHD development in nine, donor unavailability in seven, disease relapse in three, and secondary malignancy in one patient, respectively. Nine patients were still on treatment with DLI, and they received a median of four (range 2–12) doses. Fourteen percent of patients developed transient grade-II acute GVHD while 12% developed chronic GVHD post-DLI administration. Acute GVHD was managed successfully with short course steroids, and four out of five patients with cGVHD were disease-free and off immunosuppression. With a median follow-up of 44 months (range 8–120), relapse-free (RFS) and overall survival (OS) were 74%, (95% CI, 54–87%) and 78%, (95% CI, 58–89%) respectively, while the cumulative incidence of non-relapse mortality (NRM) was 13% (95% CI, 4–28%). The cumulative incidence of relapse in patients with intermediate and high DRI is 7% and 15%, respectively. Conclusion: Prolonged—up to three years—low-dose pro-DLI administered every two months is safe and effective in reducing relapse rate in patients with high-risk acute leukemia. The low-dose repetitive administration DLI strategy reduced the risk of DLI-mediated GVHD, while the prolonged repeated administration helped in preventing relapse, possibly by inducing a sustained and prolonged immunological pressure on residual leukemic cells. This novel strategy deserves testing in larger cohort of patients with high-risk acute leukemia.

## 1. Introduction

Allogeneic hematopoietic cell transplantation (allo-SCT) is currently the treatment of choice for patients with high-risk acute myeloid leukemia (AML), as well as for patients with high-risk acute lymphoblastic leukemia (ALL) and myelodysplastic syndrome (MDS) [1]. Allo-SCT therapeutic efficacy is mainly dependent on immune alloreactivity mediated by donor lymphocytes infused with the stem cell graft, the so-called graft versus leukemia (GVL) effect. The introduction of reduced intensity (RIC) and non-myeloablative (NST) conditioning regimens resulted in the extended application of allo-SCT to elderly patients as well as in those with various comorbidities who were previously considered ineligible for transplantation [2].

Disease recurrence is the major obstacle to the success of allo-SCT as patients with high-risk acute leukemia and MDS have a high risk of relapse after allo-SCT [3,4]. The outcome of patients with acute leukemia who relapse after allo-SCT is extremely poor. In a retrospective analysis conducted by CIBMTR, the overall survival (OS) at 1 year for all patients with AML who relapsed after previous allo-SCT was 23%. In multivariable analysis, variables significantly associated with decreased OS were the short time from allo-SCT to relapse, age above 40 years, active GVHD at the time of relapse, and the use of mismatched-unrelated donor and/or cord blood [4,5].

In an effort to reduce the relapse rate, various therapeutic methods have been implemented into clinical practice. Among them, prophylactic donor lymphocyte infusion (DLI) has shown significant efficacy but at the cost of increased graft versus host disease (GVHD) [6,7]. Acute leukemia is a biologically aggressive disease with a high proliferation rate, and the efficacy of therapeutic DLI administration at the time of overt hematological relapse is limited and usually of short duration [8]. Therefore, DLI is a more reasonable therapeutic approach when administered at earlier time points [9]. Pre-emptive DLI (pre-DLI) has been defined as the DLI administration to patients with persistent minimal residual disease (MRD) and/or mixed donor chimerism (MC). In contrast, DLI administered to patients without detectable MRD and in a state of complete donor chimerism (CC) has been defined as prophylactic DLI (pro-DLI) [6,10]. Previous studies in children and adult patients with AML have shown the efficacy of pre-DLI and pro-DLI in reducing the relapse rate in comparison with historical control groups [11,12].

Despite the significant heterogeneity regarding patient populations and DLI specifics such as cell dose, time from allo-SCT, frequency of administration and more, in almost all of these studies, either a single or a few doses of DLI were administered for a relatively short period of time [6,7,8,9,10,11,12]. In the case of repeated administration, DLI was usually given in an escalating manner, i.e., the cell dose was increased in each subsequent cycle in order to reduce the risk of GVHD, which is dose- and time-dependent [13,14].

From a theoretical point of view, DLI administered as a single or limited dose for a relatively short period of time might not be enough for preventing post-allo-SCT leukemic relapse, as you may need time-persistent anti-leukemic immune reactivity in order to eliminate the very last leukemic cell. The immune pressure on the leukemic clone mediated by the GVL effect is also dependent on the total number of the donor immune effectors with anti-leukemic reactivity. On the other hand, DLI administered in an escalating manner usually results in GVHD, and the subsequent necessity for immunosuppressive treatment represses the anti-leukemic immune response. With this theoretical background, we initiated the current study aiming to test the safety and efficacy of a novel method of prophylactic DLI based on prolonged repetitive administration of low lymphocyte doses. Our hypothesis was that repeated and prolonged DLI administration will result in preventing relapse by inducing a long-lasting anti-leukemic effect, while low-dose lymphocyte dosing may help in minimizing toxicity and especially DLI-associated GVHD.

## 2. Patients and Methods

### 2.1. Inclusion and Exclusion Criteria

Prophylactic DLI was administered to patients with high-risk AML or ALL. High-risk leukemia at the time of transplantation was defined as follows: (1) AML with unfavorable genetic characteristics (ELN-2017) in CR1, (2) AML with intermediate risk genetics in CR2 [15], (3) ALL patients with poor risk genetics in CR1, (4) ALL patients beyond CR1, (5) patients with refractory disease, (6) patients with measurable residual disease at the time of transplantation. In order to receive DLI patients had to be in CR without detectable MRD post-transplantation and off immunosuppression (IS).

Exclusion criteria were: (1) severe acute GVHD (grade III-IV), (2) chronic GVHD, (3) ongoing immunosuppressive therapy, (4) not achieving CR post-allo-SCT at the time of planned DLI administration, and (5) detectable MRD post-allo-SCT, as these patients have an extremely high risk of relapse and we consider them to be in need of more aggressive or experimental approaches [16].

Patients with persistent MC were eligible for prophylactic DLI if they had high-risk AML or ALL according to the above definition. Patients with persistent MC and AML or ALL not fulfilling the criteria for a high-risk definition were not included in this analysis. A percentage of the patients reported in our study have been previously presented in a manuscript, including patients from two academic centers [17]. However, the present study includes a much larger number of patients treated in a single institution, with the same protocol of DLI administration and with longer follow-up. The study was approved by the Institutional Review Board and Bioethics Committee of ATTIKO University Hospital (EBD277-Approved May/2021). Written informed consent was given by all patients and donors.

Between 2011 and 2020, one hundred and eight patients who underwent allo-SCT in our department fulfilled the proposed criteria for definition of high-risk acute leukemia and were considered as potential candidates for treatment with pro-DLI. However, 64 out of 108 did not receive pro-DLI due to a variety of reasons, such as development of GVHD, early hematological or molecular relapse, early death, severe infection, donor unavailability or patient refusal to consent. Details are shown in Figure 1.

### 2.2. DLI Administration

DLI was administered at a standard dose of 2 × 10^6^/kg CD3^+^ cells, with the first dose infused at a median of 190 days (range, +140 to +220) post-allo-SCT and repeated every two months for at least 36 months post-allo-SCT. DLI was discontinued in case of disease relapse, in the presence of clinical and/or laboratory features suggestive of GVHD and/or in the case of donor unavailability. Continuation of DLI administration on a bimonthly basis beyond the 36-month period was at the discretion of the treating physician. The source of DLI in most cases of MSD was fresh peripheral blood (PB). Original donors were checked every two months for infectious disease markers and CD3^+^ number. Based on the CD3^+^ cell count, the respective volume of PB containing 2 × 10^6^ CD3^+^ cells/kg (BW of recipient) was collected in ACD and infused fresh to the recipient. Sibling donors unwilling to donate PB every two months and unrelated donors underwent leukapheresis, lymphocytes were harvested and divided in aliquots containing 2 × 10^6^ CD3^+^ cells /kg (BW of recipient) each and stored in a LN2 tank for subsequent infusion into the recipients.

### 2.3. GVHD and Chimerism Status Monitoring

Acute GVHD diagnosis and grading was based initially on Glucksberg [18] and later on the Mount Sinai Acute GVHD International Consortium (MAGIC) criteria [19]. Chronic GVHD diagnosis and staging was based on the criteria established by the Working Group of National Institutes of Health Consensus Development Project for chronic GVHD [20]. All patients were monitored carefully for the presence of clinical or laboratory signs of GVHD in each monthly visit. Bone marrow aspiration for MRD assessment was performed every month during the first 6 months following transplantation and then every 2–3 months for a period of 3 years [21]. MRD was assessed by either multiparameter flow cytometry (MPF) or quantitative PCR (qPCR). MPF was used for MRD monitoring in the vast majority of ALL and AML cases, while qPCR was used in patients with detectable gene abnormalities such as MLL rearrangements, CBF leukemias, and NPM1 mut. Chimerism analysis was also performed every 2–3 months in whole BM cells and in PB CD3^+^ cell subpopulations. The presence of host DNA elements at percentages above or equal to 5% was defined as mixed chimerism [10].

### 2.4. Objectives and Statistical Analysis

The primary endpoint of the study was the cumulative incidence of relapse, while secondary endpoints were the cumulative incidence of acute and chronic GVHD, relapse-free (RFS) and overall survival (OS). OS was defined as the time from transplantation to last follow-up or death due to any cause, while RFS was defined as the time from transplantation to last follow-up, relapse or death due to any cause. The disease risk index (DRI) was included as a variable in statistical analysis [22]. Categorical variables between groups were compared using the two-sided Fisher’s exact test, while the nonparametric Mann-Whitney U test was used to compare continuous variables. Competing risk analysis was used for estimating the cumulative incidence of GVHD, relapse rate and non-relapse mortality (NRM). Gray’s test was used for univariate analysis of GVHD, relapse and NRM, while Fine and Gray proportional hazards regression model was used for multivariate analysis. OS and RFS were estimated by using the Kaplan-Meier method with log-rank test for univariate analysis, while Cox proportional hazards regression model with stepwise forward selection was used for multivariate analysis. Statistical analysis was performed with the use of easy R and Medcalc statistical software [23].

## 3. Results

### 3.1. Patients and DLI

Forty-four patients with a median age of 53 years (range 20–67), 24 males and 20 females, who underwent allo-SCT between 2011 and 2020 were included in the study (Table 1). Thirty-three had high-risk AML and 11 had high-risk ALL. Seventeen patients with AML and five patients with ALL had unfavorable genetic features at diagnosis. Twenty-three patients were transplanted in first CR, 12 patients in remission but beyond CR1, while nine patients underwent allo-SCT with refractory disease. Nine out of 23 (39%) patients transplanted in CR1 had detectable MRD at the time of transplantation. Disease risk index (DRI) was high and intermediate in 23 (52%) and 21 (48%) patients, respectively. Donors were matched siblings (MSD) in 38 patients and matched unrelated (MUD) in six patients. The source of stem cells was G-CSF mobilized peripheral blood (PBSC) in all cases. Myeloablative (MAC) and reduced intensity conditioning (RIC) was administered to 36 and 8 patients, respectively. GVHD prophylaxis was CSA in combination with low-dose alemtuzumab (5 mg/dose, total dose 10 mg) on days −2 and −1 in 39 patients (89%) or low-dose MTX in five patients (11%) (Table 1).

Thirty-five patients (80%) completed the scheduled treatment. Fifteen out of the 35 patients received all planned DLI doses within 36 months. DLI administration was continued after the completion of the 15 planned doses in 8 out of 15 patients. DLI was discontinued in 20 patients due to GVHD development in nine, donor unavailability in seven, disease relapse in three, and development of secondary malignancy (lung cancer) in one patient, respectively. The median number of DLI doses administered to the group of 35 patients who completed the study protocol was eight (range 1–35). Nine patients were on treatment at the end of the study and had received a median of four doses (range 2–12). The median follow-up for the whole cohort of patients was 44 months (range 8–120).

### 3.2. The Effect of DLI on Chimerism Status

Twelve out of 44 (29%) patients were in a status of MC in PB CD3^+^ cell subpopulation at the time of first DLI infusion. Seven out of these 12 patients were also in a state of low-level MC on total bone marrow (BM) cell population. At the time of first DLI dose, the median host DNA percentage was 21% (range 9–71%) and 4% (range 0–14%) in PB CD3^+^ subpopulation and total BM cells, respectively. Six out of seven patients with MC in BM cells converted to CC at a median of four months (range 2–6) after the first DLI dose. Only one out seven patients remained in MC in BM cells after nine DLI doses. Ten out 12 patients with MC in the PB CD3^+^ subpopulation converted to CC at a median of eight months (range 1–24) after the first DLI dose (Figure 2). Only one out of 12 patients remained in MC in PB CD3^+^ cells after the completion of all 15 DLI infusions, while another patient was still in a status of MC in PB CD3^+^ cells after nine DLI doses.

### 3.3. GVHD and Non-Relapse Mortality

Acute GVHD II-IV developed in 6 out of 44 patients, with a cumulative incidence of 14% (95% CI, 6–27%). The onset of aGVHD occurred at a median of two months (range 1–4) after DLI administration. Overall peak grade of aGVHD was II in all six patients. Acute GVHD of lower intestinal track was present in all six patients, while two out of six patients also had liver involvement. In all instances, aGVHD developed after the first or second DLI infusion and no case of aGVHD was observed later on, despite the continuous administration of DLI every two months for 36 months. Acute GVHD was easily managed and resolved in all cases with short courses of steroids. None of the patients developed higher grades (III-IV) of aGVHD. Further DLI administration was discontinued in the six patients who developed aGVHD.

Overall, cGVHD occurred in five out of the 44 patients, with a cumulative incidence of 12% (95% CI, 4–23%) (Figure 3). Onset of cGVHD occurred at a median of three months (range 2–8) after DLI administration. The overall grade of cGVHD was mild in three cases, while it was moderate and severe in one patient each, respectively. Notably, the cumulative incidence of severe cGVHD was extremely low and estimated as 2.3% (95% CI, 2–10%). At the conclusion of the study, four out of five patients were free of active cGVHD and off immunosuppression.

The cumulative incidence of NRM was 13% (95% CI, 4–28%). Overall, four patients died, one from respiratory infection 43 months post-allo-SCT, and one due to refractory Guillain Barre syndrome 23 months after allo-SCT. Two other patients died of secondary malignancy (lung and gastric) 21 and 46 months after allo-SCT. All four patients were free of GVHD and off immunosuppression, and the cause of death was not related to DLI administration.

### 3.4. Relapse and Overall Survival

Relapse was observed in only four out of 44 patients after 8-, 14-, 21- and 56-months post-allo-SCT. The cumulative incidence of relapse was 12% (95% CI, 4–27%) (Figure 4). Specifically, relapse occurred in four patients with high-risk AML. Three out of four patients had poor risk cytogenetic features, including complex monosomic karyotype and MLL rearrangement other than t (9; 11). All three patients developed isolated extramedullary relapse. Another patient with t (8; 21) who underwent allo-SCT at a state of refractory disease had a late BM relapse 56-months post-transplant. Of note, this patient developed donor cell leukemia with distinct molecular abnormalities (IDH1-mut and RAS-mut) not present at initial diagnosis. The cumulative incidence of relapse in patients with intermediate and high DRI was 7% (95% CI, 1–27%) and 15% (95% CI, 4–33%), respectively.

With a median follow-up of 44 months (range 8–120), the RFS and OS were 74% (95% CI, 54–87%) and 78% (95% CI, 58–89%), respectively (Figure 5).

## 4. Discussion

In the current study we demonstrated that prolonged (up to three years) low dose prophylactic-DLI administered every two months is safe and effective in reducing relapse rate in patients with high-risk acute leukemia. The strategy of repetitive low-dose DLI administration reduces the risk of DLI-mediated aGVHD and cGVHD to 14% and 12%, respectively, while it helps in preventing relapse by reducing its frequency to only 13%, possibly by inducing a sustained immunological pressure on the leukemic cells. 

The beneficial effect of pro-DLI in prevention of relapse has been reported in several previous phase I and II studies. However, these studies are characterized by a significant heterogeneity in terms of patient populations and transplant modalities. Moreover, in most of these studies, an average of two–three pro-DLI doses were usually administered in an escalating manner. In the absence of GVHD, DLI was repeated using incremental cell dosing at regular intervals [24,25,26,27].

The escalating strategy of pro-DLI administration is an effective method for prevention of relapse in high-risk leukemia but at the cost of increased severe GVHD. Pro-DLI was an integral part of the FLAMSA protocol pioneered by Kolb et al. In their prospective study of 75 patients with high-risk MDS or AML, pro-DLI was administered in patients free of GVHD after withdrawal of IS. Pro-DLI was administered with an escalating manner, starting with a dose of 2 × 10^5^ and 1 × 10^6^/kg CD3^+^ cells in MUD and MSD transplantations, respectively. In the absence of GVHD, escalating CD3^+^ cell doses were administered at 4–6-week intervals. Pro-DLI resulted in improved outcomes, with a three-year OS of 42%, with GVHD being the main complication. Acute and chronic GVHD occurred in 61% and 45% of the patients, while the incidence of severe grade III-IV aGVHD was 24%. The benefit in terms of OS was most significant in patients who developed limited chronic GvHD [28].

In terms of safety, previous studies have shown that the most important parameters associated with the development of DLI-associated GVHD are the time from transplant to first DLI dose and the CD3^+^ cell dose administered with each DLI dose. Despite the significant heterogeneity between studies, a common observation that emerged was that DLI administration before day +100 is associated with an unacceptably high rate of severe GVHD [29,30,31]. In fact, the longer the duration between transplantation and the first DLI dose, the less the possibility of GVHD development. Administration of DLI at earlier time points, before the repair of tissue damage induced by the conditioning regimen and the associated cytokine storm, results in stronger alloimmune responses and significant tissue inflammation manifesting as severe GVHD [32]. The efficacy and safety of delayed DLI has been tested in an experimental mice model. Similar to human data, the results from experimental animal models suggest that delayed DLI administration results in significantly reduced GVHD without loss of the beneficial GVL effect [33].

In a large EBMT registry-based study, the median time from transplant to relapse was six months, and therefore we assume that DLI administration beyond six months from transplantation will be of limited value in prevention of a high proportion of relapses [4]. Based on the above observations, in our study we chose to administer the first dose of pro-DLI at a median of six months post-transplant, with the aim to avoid severe GVHD on the one hand and to have a substantial effect on relapse prevention on the other hand. 

Regarding the CD3^+^ cell dose, previous studies have shown that the more the CD3^+^ cell number, the higher the possibility of developing severe GVHD. In the setting of MSD and a well-matched unrelated donor (WMUD), a single CD3^+^ cell dose of 1–2 × 10^6^/kg is acceptable since it rarely induces severe GVHD [10]. A previous study including 100 patients with various hematological malignancies examined the effect of therapeutic DLI administration on the outcome of post-transplant relapse. The vast majority of the patients received DLI with an escalating-dose regimen. The CD3^+^ cell dose administered per infusion ranged from 5 × 10^7^/kg to 2 × 10^8^/kg, and the median total number of CD3^+^ cells infused was 7.6 × 10^7^/kg (range 0.01–25.2). The median number of DLI doses per patient was 4.5 (range 1–11), while the median duration of DLI courses was 84 days (range 1–900). The incidence and severity of GVHD was associated with the number of CD3^+^ cells administered in each single infusion, while the total CD3^+^ cell dose infused had no effect on GVHD. Moreover, in univariate analysis, the time interval between DLI infusions was inversely associated with the incidence and severity of GVHD. In more detail, the shorter the time interval between DLI doses, the higher the incidence of GVHD. Parameters associated with the efficacy of DLI were type of disease and status of disease at the time of relapse, with chronic phase CML at molecular or hematological relapse having the highest probabilities for a successful outcome. The CD3^+^ cell dose had no effect on the response rate after DLI administration [34].

In a similar study, the safety and efficacy of therapeutic DLI was examined in 344 patients with CML relapse after allo-SCT. Based on the initial mononuclear cell (MNC) dose infused, patients were divided into three groups (98 patients in group A received less than 0.21 × 10^8^/kg, 107 patients in group B received between 0.21 to 2.0 × 10^8^/kg, and 93 patients in group C received more than 2.0 × 10^8^/kg). Two or more DLI doses were administered in 103 patients, with additional DLI infusions given at a median interval of 44 days. In multivariate analysis, the initial DLI cell dose was statistically associated with the incidence and severity of GVHD- and DLI-related mortality but not with the response to DLI. The effect of the total DLI dose as well as the time interval between DLI doses on the safety and efficacy of DLI was not examined in this study because only a minority of patients received more than two DLI doses [35].

Our study included only patients matched at the allele level at 10 loci with either related or unrelated donors. The CD3^+^ cell dose infused with each DLI administration was 2 × 10^6^/kg and repeated every two months for at least 36 months or until relapse or GVHD development, whichever occurred first. Despite the median total CD3^+^ cell dose infused being 1.6 × 10^7^/kg, the incidence of GVHD was extremely low and easily manageable because, as in previous studies, the total CD3^+^ cell dose had no effect on GVHD incidence and severity. Indeed, in our study none of our patients developed acute GVHD after the third DLI dose and despite the continuation of DLI infusions, as a further proof of the concept that the total CD3^+^ cell dose is not associated with GVHD incidence. 

Taken together, the data from our study as well as from previous studies show that the CD3^+^ cell dose infused with a single DLI dose and the time interval between subsequent doses were the most significant parameters associated with DLI-induced GVHD. The escalating strategy of DLI administration, although effective for prevention of relapse in high-risk leukemia, is often associated with significant DLI-induced severe GVHD. On the contrary, the total CD3^+^ cell dose was not associated with GVHD incidence. In our study, we showed that in the setting of well-matched related and unrelated donors, the CD3^+^ cell dose of 2 × 10^6^/kg administered every two months for at least 36 months is associated with a significantly low possibility for GVHD development.

From a theoretical point of view, a major limitation of single-dose DLI administration is that the strength of the induced alloreactivity tends to decrease over time as a result of the development of mutual immune tolerance that often occurs in allogeneic chimeras, especially in the absence of GVHD. Therefore, it is reasonable to assume that the anti-leukemic activity of a single DLI administration occurs for a limited period time. In order to achieve a sustained immunological pressure on residual leukemia, DLI administration should repeated at regular intervals for a prolonged period of time post-transplantation.

The concept of multiple DLI doses infused over time has been tested only in a few previous studies. A different approach than in our protocol was explored by Chinese researchers. In this study, a total of 100 patients with AML and ALL received multiple pro-DLI infusions, based on the presence of positive MRD and active GVHD, for prevention of relapse post-allo-SCT. The first pro-DLI was administered at day +30 and day +45 to +60 in patients after MSD and mismatched unrelated or haploidentical allo-SCT, respectively. The median dose of CD3^+^ cells in each DLI was 3.7 × 10^7^/kg (range, 1.8–6.6). Low-dose CyA was administered after each DLI infusion in order to prevent GVHD. Pro-DLI was repeated if MRD was positive and GVHD was absent. In more detail, patients with negative MRD and absent or resolved GVHD received a second DLI dose six months after the first dose, while patients with positive MRD received DLI every three months for a period of 12 months or until GVHD development. Despite the high-risk disease status, pro-DLI resulted in better leukemia control with a three-year cumulative incidence of relapse of 32% and LFS and OS of 50% and 51%, respectively. Multivariate analysis revealed that MRD positivity after the first DLI dose and single pro-DLI versus 2–4 DLI doses were the only parameters statistically associated with increased incidence of relapse, indicating the beneficial effect of multiple DLI infusions in preventing relapse after allo-SCT. However, different from our study, the CD3^+^ cell dose in each DLI was relatively high and administered very early in the post-transplant course, resulting in increased incidence of 38% acute GVHD grade II-IV and 61% moderate chronic GVHD [24].

In our study, low-dose alemtuzumab was used for GVHD prophylaxis in 89% of patients. Although its half-life in stem cell transplant recipients ranges between 8 and 21 days, alemtuzumab may have an impact on the safety and efficacy of our repeated prolonged low-dose DLI strategy [36].

In a study by Finke et al., the outcome after DLI administration was examined in a cohort of 93 high-risk AML patients. Similar to our study, a significant percentage of patients received alemtuzumab during conditioning for GVHD prophylaxis. DLI was administered as a treatment of post-transplant relapse in 51 patients, while 42 patients received DLI preemptively for persistent MC or detectable MRD. The median number of pre-DLI doses administered was 6 (range 1–43), the median CD3^+^ cell dose was 2.3 × 10^6^/kg (range 0.1–6.8) and median time from allo-SCT to first dose of pre-DLI was 273 days, (range 80–2146). Different than our study, in the Finke study pre-DLI doses were not infused at regular time intervals and they were administered for a shorter period of time. The median time between first and second pre-DLI dose was 28 days (range 14–1778) and last pre-DLI dose was administered at a median of 161 days (range 53–2114) after the first dose. The incidence of acute and chronic GVHD following pre-DLI was 29% and 14%, respectively. The median five-year OS for the whole cohort of preemptively treated patients was 43%. The outcome was better in patients who received pre-DLI due to persistent MC (5-year OS of 58%) compared to those receiving DLI for treatment of detectable MRD (5-year OS of 19%). A significant percentage of patients (31%) underwent a second allo-SCT for treatment of relapse, while relapse was the cause of death in 21% of patients [16].

An additional advantage of our study is that it included a homogenous group of patients treated with the same protocol in a single center. As the vast majority of our patients received low-dose alemtuzumab for GVHD prevention, we cannot exclude that alemtuzumab may have contributed to the success of our prolonged, repeatedly low-dose prophylactic-DLI approach and thus it will be interesting to learn if the successful results can be achieved in other patient cohorts receiving different GVHD prophylaxis. One of our study limitations is the rather small number of patients, which did not allow for meaningful comparisons between subgroups of patients with different disease characteristics.

To summarize, in the current study we examined the efficacy and safety of a novel strategy based on multiple repeated doses of low-dose pro-DLI administered for 36-months post-allo-SCT for prevention of relapse in patients with high-risk acute leukemia. DLI was administered at the same CD3^+^ cell dose every eight weeks until the completion after three years. By this innovative approach we were able to meaningfully reduce the relapse rates to 7% and 15% in patients with intermediate and high DRI, respectively, while the incidence of severe cGVHD after pro-DLI remained as low as 2.4%.

## 5. Conclusions

In conclusion, in our study we observed that prophylactic-DLI is safe and effective in reducing the relapse rate of patients with high-risk acute leukemia. The novelty of our method is based on the repeated low-dose dosing. It is possible that by administering low doses severe GVHD is avoided, while the repeated administration helps in preventing relapse by inducing a sustained and prolonged immunologic pressure on residual leukemic cells that survived the conditioning regimen. We consider that our strategy deserves testing in a larger cohort of patients.

## Figures and Tables

**Figure 1 cancers-13-02699-f001:**
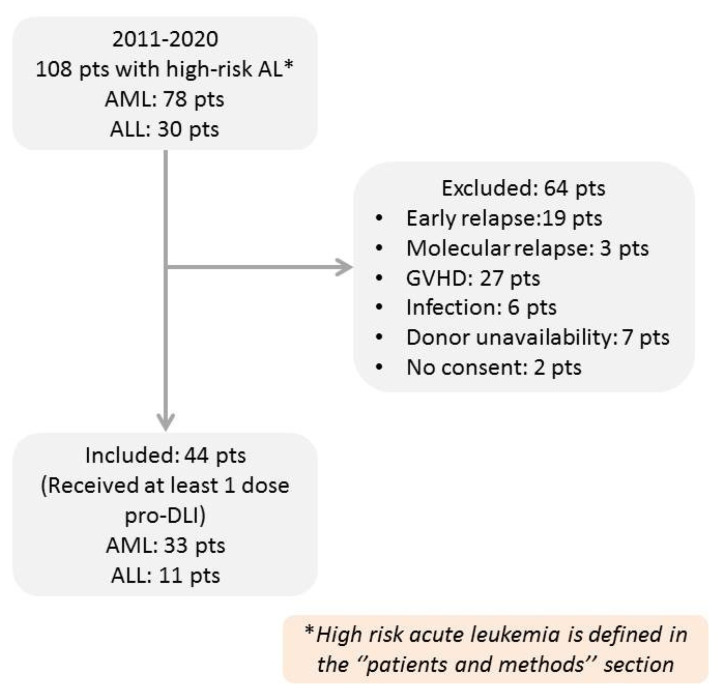
Treatment flow diagram.

**Figure 2 cancers-13-02699-f002:**
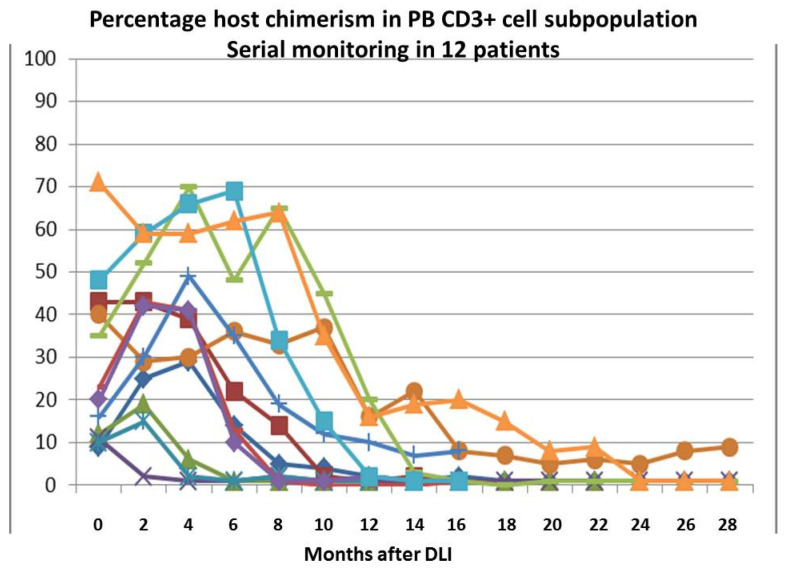
Percentage of host chimerism in peripheral blood CD3^+^ lymphocytes: Serial monitoring in 12 patients.

**Figure 3 cancers-13-02699-f003:**
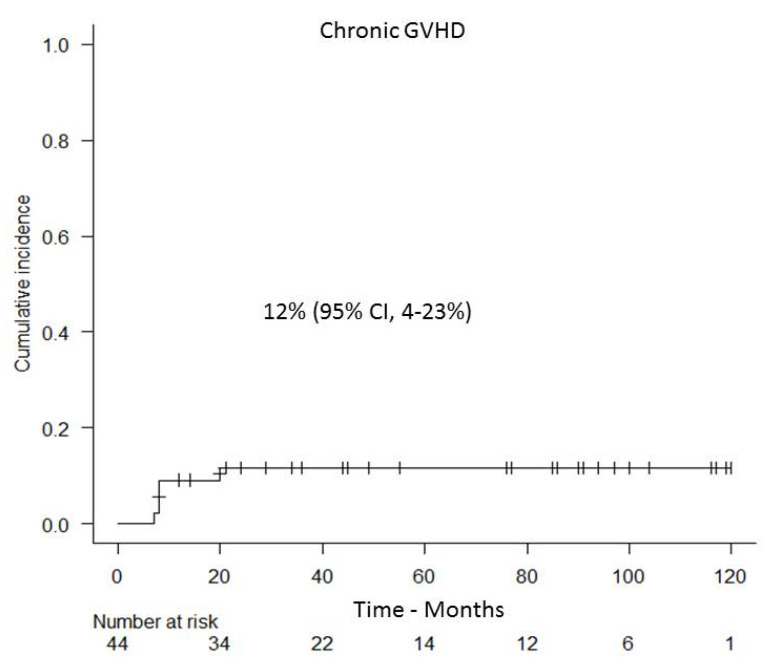
Cumulative incidence of chronic GVHD.

**Figure 4 cancers-13-02699-f004:**
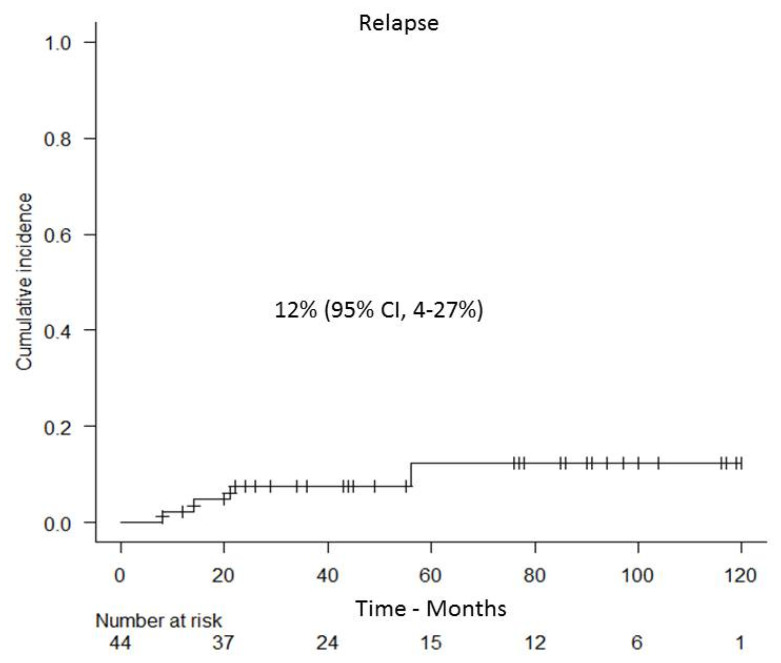
Cumulative incidence of relapse.

**Figure 5 cancers-13-02699-f005:**
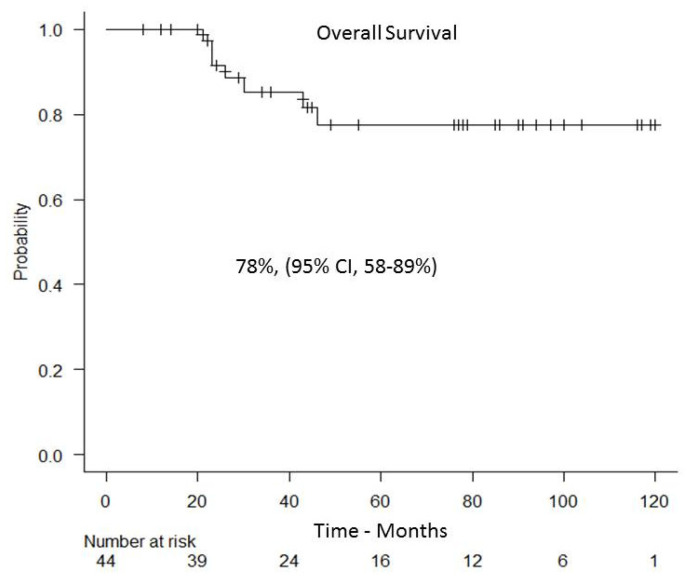
Overall survival.

**Table 1 cancers-13-02699-t001:** Patient characteristics (No = 44).

Age, Median, (Range)	53 Years (20–67)
**Sex, Male/Female**	24/20
**Acute Myeloid Leukemia** Good risk genetics * Intermediate risk genetics * Poor risk genetics * CR1 ≥CR2 Refractory disease MRD positive **	33 1/33 15/33 17/33 17/33 8/33 8/33 4/33
**Acute Lymphoblastic Leukemia** Standard risk Poor risk CR1 ≥CR2 Refractory disease MRD positive **	11 6/11 5/11 6/11 4/11 1/11 5/11
**Disease Risk Index** Intermediate High	21 23
**Donor** Matched related donor Matched unrelated donor	38 6
**Conditioning regimen** Myeloablative Reduced intensity	36 8
**GVHD prophylaxis** Cyclosporine plus low dose Alemtuzumab Cyclosporine plus short course MTX	39 5

* ELN-2017, ** patients in CR1 at the time of transplantation.

## Data Availability

Data is contained within the article.

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
