# Peer review of "Repetitively Administered Low-Dose Donor Lymphocyte Infusion for Prevention of Relapse after Allogeneic Stem Cell Transplantation in Patients with High-Risk Acute Leukemia"

_cancers, 2021, doi:10.3390/cancers13112699_

Round 1

Reviewer 1 Report

A good and important paper.

Minor suggestions:

  • Page 4, the last but one line. Clarify dose of alemtuzumab: 5 mg per dose (5mg twice) or 5 mg total (2.5mg twice).
  • Page 9, the last but one paragraph, the first sentence: “vast majority” after “the median time” is not correct by definition: it will be exactly ½ of patients. I suggest changing it for “high proportion”, “approximately a half” or similar.

Author Response

Below there is our response to comments raised by 1st reviewer 

Minor suggestions:

Point 1: Page 4, the last but one line. Clarify dose of alemtuzumab: 5 mg per dose (5mg twice) or 5 mg total (2.5mg twice).

Response to point 1: The dose of alemtuzumab was 10mg total (5mg daily , days -2, and -1). This issue has been clarified in the revised version of our manuscript. (See yellow marked)

Point 2: Page 9, the last but one paragraph, the first sentence: “vast majority” after “the median time” is not correct by definition: it will be exactly ½ of patients. I suggest changing it for “high proportion”, “approximately a half” or similar.

Response to point 2: The sentence has been corrected in the revised version of our manuscript and the new sentence is as follows: ‘’ In a large EBMT registry based study, the median time from transplant to relapse was 6 months, and therefore delayed DLI administration beyond this time will be of limited value for the prevention of a high proportion of relapses’’. (See yellow marked)

Reviewer 2 Report

Tsirigotis et al. reported a study of injection of a fixed low dose of donor lymphocytes following allogeneic transplantation in patients with AML and ALL. The infusion was started at a median of 6 months post-transplant.

I have few comments or questions:

  1. It was not clear to me by reading the text at the beginning whether the study was prospective. This should be clearly stated as usual (inclusion criteria, primary objectives, secondary objectives, registered nber)
  2. In this study reported, all patients received at least one DLI. It would be important to precise the number of patients who were scheduled to receive the pre or pro-DLI but could not receive it because of GVHD, early relapse or severe infection. I presume that the patients were pre-enrolled before transplantation with definitive inclusion following transplantation. The number of patients pre-enrolled would be of interest to know.
  3. The first dose of DLI was given at a median time of 190 days (~6months). Since the majority of patients with AML relapse during the first 6 months, the benefit of late DLI should be considered in the discussion.
  4. In the introduction, preemptive-DLI (pre-DLI) is defined as DLI to patients with persistent MRD and/or mixed chimerism, while prophylactic-DLI is defined as DLI to patients without detectable MRD and complete chimerism. Unless, I’m mistaken, no MRD data post-transplant are reported. This should be described. At what time the MRD post-transplant was examined and which technique were used to detected in the patients with AML ?
  5. In the same line of thought, how many patients were in the pre-DLI situation versus pro-DLI ? Were there any differences in the outcome?
  6. Details of the patients who relapsed (diagnosis (AML? ALL ?), cytogenetics characteristics) might be mentioned.
  7. Page 7 : 1st § sentence “acute GVHD was grade II in all 6 patients involving the intestine in 4 and liver in other 2 patients” is unclear: since it is written earlier that 6 pts had grade II-IV GVHD.
  8. References 10 and 35 are the same

Author Response

Below there is our response to comments raised by 2nd REVIEWER

I have few comments or questions:

Point 1: It was not clear to me by reading the text at the beginning whether the study was prospective. This should be clearly stated as usual (inclusion criteria, primary objectives, secondary objectives, registered nber)

Response to point 1: The study was prospective since it reflects our standard policy for the treatment of our patients with acute leukemia in our department during the last decade. Study protocol, received approval by our institutional review board and bioethical committee of ATTIKON General University Hospital. The local registration number given by ATTIKON IRB is ‘’ΕΒΔ277’’. This is stated in subsection [2.1. Inclusion and exclusion criteria] and marked in yellow

Inclusion and exclusion criteria are presented in detail in subsection [2.1. Inclusion and exclusion criteria]    

The primary endpoint of the study was the cumulative incidence of relapse while secondary end points were cumulative incidence of acute and chronic GVHD, relapse free and overall survival. Study endpoints are presented in subsection [2.4. Objectives and Statistical analysis] marked in yellow    

Point 2: In this study reported, all patients received at least one DLI. It would be important to precise the number of patients who were scheduled to receive the pre or pro-DLI but could not receive it because of GVHD, early relapse or severe infection. I presume that the patients were pre-enrolled before transplantation with definitive inclusion following transplantation. The number of patients pre-enrolled would be of interest to know.

Response to point 2: These data are presented in the revised version of our manuscript. Actually between 2011 and 2020, 108 patients who underwent allo-SCT in our department fulfilled the criteria for definition of high risk acute leukemia and therefore were potential candidates for treatment with pro-DLI. However, only 44 patients received pro-DLI, while 64 patients did not receive pro-DLI due to various reasons, such as GVHD, relapse, infection, etc. These data are included in subsection [2.1. Inclusion and exclusion criteria] and marked in yellow. Also the data are presented in Figure 1.    

Point 3: The first dose of DLI was given at a median time of 190 days (~6months). Since the majority of patients with AML relapse during the first 6 months, the benefit of late DLI should be considered in the discussion.

Response to point 3: We absolutely agree with the comment raised by the reviewer. Indeed in Discussion we provide data that the median time from transplant to relapse is 6 months, and therefore the more the delay beyond this time the less the efficacy in prevention of relapse. In other words, administration of pro-DLI long enough after the sixth month will be of limited efficacy since many of the relapses occur before this time. On the other hand, administration of pro-DLI before 6 months increases the incidence of GVHD. Therefore in our study we choose to administer the first dose of pro-DLI after 6 months from the time of transplant. See paragraph in yellow marked in discussion    

Point 4: In the introduction, preemptive-DLI (pre-DLI) is defined as DLI to patients with persistent MRD and/or mixed chimerism, while prophylactic-DLI is defined as DLI to patients without detectable MRD and complete chimerism. Unless, I’m mistaken, no MRD data post-transplant are reported. This should be described. At what time the MRD post-transplant was examined and which technique were used to detected in the patients with AML ?

Response to point 4: Indeed in subsection [2.3. GVHD and chimerism status monitoring] it is written that MRD was performed by either multiparameter flow cytometry (MPF) or quantitative PCR. MPF was the method used in the vast majority of cases, while PCR was used in cases with detectable gene abnormalities as stated in subsection in yellow marked. Also by definition patients with detectable MRD after transplant and before the first pro-DLI administration were not included in the study. These patients have a high probability for relapse and are not treated with pro-DLI. Instead these patients received other treatments such as higher doses of DLI in combination with azacytidine in cases with AML. Only patients with negative MRD were included in our study. Also this is stated in subsection [2.1. Inclusion and exclusion criteria] and marked in yellow.   

Point 5: In the same line of thought, how many patients were in the pre-DLI situation versus pro-DLI ? Were there any differences in the outcome?

Response to point 5: Indeed these details are presented in subsection [3.2. The effect of DLI on chimerism status]. Twelve out of 44 patients were in a state of mixed chimerism at the time of first pr-DLI administration. However MC was not the only reason for DLI administration to these patients. All these 12 patients had high risk acute leukemia and therefore they were selected for pro-DLI administration. The main reason for administering DLI was the disease risk status for relapse. Patients with MC and without high risk acute leukemia were not included in our study. Regarding the outcome we consider that the number of patients is not large enough for performing statistical analysis and reach to meaningful conclusions.  

Point 6: Details of the patients who relapsed (diagnosis (AML? ALL ?), cytogenetics characteristics) might be mentioned.

Response to point 6: The type of disease, the type of relapse and cytogenetic abnormalities are presented in subsection [3.4. Relapse and Overall Survival] in the revised version of our manuscript. See in marked yellow

Point 7: Page 7 : 1st § sentence “acute GVHD was grade II in all 6 patients involving the intestine in 4 and liver in other 2 patients” is unclear: since it is written earlier that 6 pts had grade II-IV GVHD.

Response to point 7: We absolutely agree with the comment. Indeed, this has been corrected in the revised version of our manuscript. The correct is that all 6 patients had overall peak AGVHD grade of 2. Intestine was involved in all cases while, in 2 patients liver was also involved. This is presented in a clear way in subsection [3.3. GVHD and Non Relapse Mortality] and marked in yellow.

Point 8: References 10 and 35 are the same

Response to point 8: We absolutely agree with the comment. This errotrhas been corrected in the revised version of our manuscript.

Round 2

Reviewer 2 Report

The authors answered appropriately to questions and comments